# Oral SERD, a Novel Endocrine Therapy for Estrogen Receptor-Positive Breast Cancer

**DOI:** 10.3390/cancers16030619

**Published:** 2024-01-31

**Authors:** Niraj Neupane, Sawyer Bawek, Sayuri Gurusinghe, Elham Moases Ghaffary, Omid Mirmosayyeb, Sangharsha Thapa, Carla Falkson, Ruth O’Regan, Ajay Dhakal

**Affiliations:** 1Department of Internal Medicine, Rochester General Hospital, Rochester, NY 14621, USA; niraj.neupane@rochesterregional.org; 2Department of Internal Medicine, University at Buffalo, Buffalo, NY 14203, USA; sbawek@buffalo.edu (S.B.); gurusing@buffalo.edu (S.G.); 3Isfahan Neurosciences Research Center, Isfahan University of Medical Sciences, Isfahan 8174673461, Iran; elham.moases@outlook.com (E.M.G.); omidmirm@buffalo.edu (O.M.); 4Westchester Medical Center, New York Medical College, Valhalla, NY 10595, USA; sangharsha.thapa@wmchealth.org; 5Department of Medicine, University of Rochester Medical Center, Rochester, NY 14642, USA; carla_falkson@urmc.rochester.edu (C.F.); ruth_oregan@urmc.rochester.edu (R.O.)

**Keywords:** breast cancer, estrogen receptor-positive, endocrine therapies, resistance, selective estrogen receptor degraders, SERDs, oral

## Abstract

**Simple Summary:**

Breast cancer is a common type of cancer among women. One type, estrogen receptor-positive (ER+), is treated with endocrine therapies. However, some patients develop resistance to these therapies, which is a challenge. Scientists have developed second-generation drugs called selective estrogen receptor degraders (SERDs) that can overcome the limitations of the existing treatment. These drugs are taken orally, which is more convenient for patients. SERDS are important because they offer a more effective and less invasive treatment option for patients with ER+ breast cancer who develop resistance to endocrine therapies. Several oral SERDs are currently in clinical trials, which means they are being tested on patients. If they are proven effective, they could become a standard treatment for ER+ breast cancer in the future.

**Abstract:**

Breast cancer is the most common cancer among women worldwide, and estrogen receptor-positive (ER+) breast cancer accounts for a significant proportion of cases. While various treatments are available, endocrine therapies are often the first-line treatment for this type of breast cancer. However, the development of drug resistance poses a significant challenge in managing this disease. *ESR1* mutations have been identified as a common mechanism of endocrine therapy resistance in ER+ breast cancer. The first-generation selective estrogen receptor degrader (SERD) fulvestrant has shown some activity against *ESR1* mutant tumors. However, due to its poor bioavailability and need for intramuscular injection, it may not be the optimal therapy for patients. Second-generation SERDs were developed to overcome these limitations. These newer drugs have improved oral bioavailability and pharmacokinetics, making them more convenient and effective for patients. Several oral SERDs are now in phase III trials for early and advanced ER+ breast cancer. This review summarizes the background of oral SERD development, the current status, and future perspectives.

## 1. Introduction

Breast cancer is a leading cause of cancer-related deaths among women worldwide, with over 2 million new cases diagnosed each year [1]. It is estimated that there were 297,790 new breast cancer cases in 2023, accounting for 15% of new cancer diagnoses, with 43,700 deaths [2]. Most breast cancers express the estrogen receptor (80%) [3]. The estrogen receptor (ER) plays a crucial role in the nucleus and the cytoplasm, where it functions as a post-transcriptional regulator. The estrogen receptor is important in cellular metabolism’s transcriptional and post-transcriptional regulation. Endocrine therapy (ET) has been a mainstay of breast cancer treatment for many years. ET blocks the estrogen receptor pathway and inhibits the growth and proliferation of estrogen receptor-positive (ER+) breast cancer [3]. Aromatase inhibitors (AI) and a selective ER modulator, tamoxifen, have been used to treat ER+ breast cancers. However, the development of resistance to these ETs has limited their effectiveness, driving the exploration of new treatment options [4]. Fulvestrant, a selective estrogen receptor degrader (SERD), represents a newer class of ET that targets explicitly and degrades the ER, reducing the ER’s activity and inhibiting the growth of ER+ breast cancer [5]. SERDs are a drug class that targets estrogen receptor alpha (ERα) for proteasome-dependent degradation. SERDs can induce receptor degradation by creating an unstable protein complex, which may overcome resistance mechanisms to aromatase inhibitors and selective estrogen receptor modulators (SERMs). SERDs target ERα for proteasome-dependent degradation, potentially impacting ERα heterodimerization with other proteins [6].

This unique mechanism of action offered the potential for fulvestrant to effectively treat breast cancers that have developed resistance to traditional ET [7]. Fulvestrant has been shown to prolong progression-free survival and possibly overall survival compared to AI among metastatic ER+ HER2 non-amplified (HER2−) breast cancer patients [8,9,10]. In addition, fulvestrant has been well tolerated, with the most common side effects being mild to moderate, including hot flashes, fatigue, and joint pain in clinical trials [10]. 

Our review aims to provide a comprehensive analysis of SERDs in treating estrogen receptor-positive breast cancer. We focus on elucidating the molecular mechanisms of SERDs, assessing their clinical efficacy in various therapeutic contexts, and identifying the challenges in their development and implementation. This review focuses on synthesizing current research and advancements, offering insights into the potential of oral SERDs as an effective treatment option.

## 2. Mechanism of Action of SERDs

The activity of the ER is primarily mediated through its interaction with specific DNA sequences known as estrogen response elements (EREs). The estrogen binding to the ER promotes the dimerization of the receptor and its subsequent binding to the EREs located in the promoter regions of estrogen-responsive genes. Once bound to the ERE, the ER acts as a transcription factor, recruiting coactivators or corepressors to initiate the transcription of the target genes [9]. AIs primarily act by reducing the estrogen levels available for binding to the ER, thereby decreasing its activity as a transcription factor. In contrast, SERMs bind to the ER and modulate its activity in a tissue-specific manner. For example, in breast tissue, SERMs such as tamoxifen act as antagonists, competing with estrogen for the ER binding site and preventing the activation of genes involved in cell growth and proliferation. Selective estrogen receptor degraders (SERDs) are a newer class of ET that bind to the ER and induce its degradation, thereby reducing the overall activity of the receptor. Unlike SERMs, which can act as either agonists or antagonists, SERDs act exclusively as antagonists of the ER. This class of drugs includes fulvestrant, the first and, until January 2023, the only FDA-approved SERD. SERDs can block the effects of estrogen on breast cancer cells by binding to the ER and promoting its degradation, leading to a decrease in the expression and activity of ER target genes [11] (Figure 1). Fulvestrant has been shown to induce the proteasomal degradation of the ER in a dose-dependent manner [12]. Proteasomal degradation of an ER is achieved through its ability to promote the dissociation of chaperone proteins such as heat shock protein 90 (Hsp90) and p23 from the ER, resulting in the exposure of a hydrophobic surface on the receptor recognized by the E3 ubiquitin ligase complex. The E3 ligase then adds ubiquitin molecules to the exposed lysine residues on the ER, targeting it for degradation by the proteasome [13]. Additionally, fulvestrant has been shown to have a longer duration of action than other ER antagonists due to its ability to induce downregulation and degradation of the receptor, as opposed to simply blocking its activity [11]. 

Overall, the unique molecular mechanism of the fulvestrant as a SERD has led to its successful use in treating advanced ER+ breast cancer, both as a first-line therapy and in patients who have progressed on prior endocrine therapies as a single agent or in combination with other ETs or targeted therapies [9,14,15,16,17]. 

However, fulvestrant has several limitations. Fulvestrant is only available as an injectable formulation, which can be inconvenient for patients who prefer oral medications or have difficulty with injections [18]. Compared to an AI, fulvestrant seems to be superior in treating ER+ metastatic breast cancer with an *ESR1* mutation. *ESR1* mutations play a crucial role in the development of endocrine therapy resistance in breast cancer. *ESR* mutations are expressed in the majority of breast cancers and are regulators of breast cancer development and progression. *ESR1* mutations, particularly ones in the ligand binding domain, have accounted for acquired endocrine resistance in a significant fraction of patients with metastatic disease [19]. However, fulvestrant’s efficacy against *ESR1* mutant metastatic breast cancer in the second line is modest, with median PFS ranging from 3 to 4 months [15,20]. Additionally and most importantly, after adopting CDK4/6 inhibitors in combination with AI as the standard first-line treatment for metastatic ER+ breast cancer a few years ago, newer clinical data have revealed minimal efficacy of fulvestrant monotherapy in the second line among these CDK4/6 inhibitor-treated tumors [21]. Several oral SERDs are being developed to address these drawbacks and are currently in clinical trials. 

Furthermore, research indicates that fulvestrant-resistant breast cancer cells can proliferate independently of estrogen signaling and display downregulated estrogen signaling [22,23]. This information is crucial to guiding clinical application and future research in this area, as it provides insights into the drug’s mechanism of action and resistance patterns in breast cancer cells.

Studies show that fulvestrant’s primary anticancer effect is not through ER degradation but rather through the transient immobilization of ER receptors in the nucleus [24]. A SUMOylation process takes place in this action, leading to transcriptional disturbances and the elimination of receptors [25,26,27]. Studies have shown that fulvestrant induces transient binding of ERα to DNA, followed by rapid release without loss of nuclear localization. Unlike proteasome inhibition, this mechanism occurs before receptor degradation [25]. With fulvestrant, chromatin accessibility is reduced, suggesting that chromatin remodeling at ERα target regions prevents transcription despite receptor binding. The SUMO2/3 marks were also found on chromatin in cells treated with fulvestrant but not SERMs [25,28].

As a result of further study, structural modifications to the side chain of fulvestrant have been found to eliminate the degradation abilities of the agent yet do not significantly affect its anticancer potential. In this manner, we can infer that fulvestrant has a complex interplay between the molecular actions it exerts against ER-positive breast cancer, including SUMOylation, which is responsible for its overall effectiveness in combating this disease [24,29].

## 3. Elacestrant

Elacestrant is a nonsteroidal oral SERD that degrades ERs and inhibits gene transcription, induction, and cell proliferation, specifically in ER+ breast cancer cell lines [14]. In January 2023, this drug became the first oral SERD to receive the FDA’s approval in treating ER+ HER2-negative (HER2−) metastatic breast cancer [30]. 

RAD1901-005 was a phase I study that studied the effects of elacestrant as a monotherapy among pretreated postmenopausal patients with ER+ HER2− metastatic breast cancer. Elacestrant was associated with an objective response rate (ORR) of 19.4% among patients who received RP2D, 15% in patients with prior SERD use, 16.7% in patients with prior CDK4/6i use, and 33.3% in patients with the *ESR1* mutation. The clinical benefit rate was 42.6% overall. No dose-limiting toxicity (DLT) was observed. The most common adverse effects (AEs) include nausea (33%), increased blood triglycerides (25%), and decreased blood phosphorus (25%). Lastly, most AEs were grades 1–2. The recommended phase II dose was 400 mg once daily [14]. RAD1901-106 was a phase Ib open-label non-randomized study investigating the effects of elacestrant monotherapy on the ER binding sites in 16 patients with pretreated postmenopausal ER+ metastatic breast cancer. The median reduction in tumor fluoroestradiol (FES) uptake was 89.1%. ORR was 11.1%, and the clinical benefit rate was 30.8%. Common AEs include nausea (68.8%), fatigue (50%), and dyspepsia (43.8%). Most AEs were grade 2 in severity. Findings showed reduced ER availability with elacestrant doses of 200 or 400 mg daily and a modest antitumor activity in this heavily pretreated population [31]. SOLTI-1905 ELIPSE was a phase I study investigating the effect of preoperative elacestrant monotherapy on cancer cell proliferation in postmenopausal women with treatment-naïve ER+ HER2− early stage breast cancer [32]. The primary endpoint was complete cell-cycle arrest (CCCA), defined as Ki-67 ≤ 2.7%. After four weeks, CCCA was achieved in 27% of patients. There was also a 41% relative reduction in Ki-67 from baseline. One treatment-related AE required the patient to discontinue treatment. Most AEs were hot flashes and constipation, which were all grade 1. These findings showed that elacestrant was associated with good biological response with a tolerable safety profile [33]. 

The encouraging data from the RAD1901-005 study supported the development of the EMERALD trial, an active phase III trial comparing the efficacy and safety of elacestrant with the standard of care (SOC) ET (fulvestrant or exemestane) in patients with breast cancer who have previously received a CDK4/6 inhibitor [14]. The EMERALD trial showed an increase in progression-free survival (PFS) (6 months PFS 34.3% (95% CI, 27.2 to 41.5) versus 20.4% (95% CI, 14.1 to 26.7), HR 0.70 (0.55 to 0.88)) associated with elacestrant versus SOC. Among the patients with the *ESR1* mutation, the 6-month PFS was 40.8 vs. 19.1 months (HR 0.55 (0.39 to 0.77)), demonstrating more benefit with elacestrant vs. SOC in this cohort than the benefit seen in all patients. Among patients without *ESR1* mutant tumors, the 6-month PFS was 28.58 vs. 21.85 months (HR 0.866 (0.628 to 1.186)), numerically showing the trend toward favoring elacestrant but not reaching statistical significance. Twenty-seven percent of patients on the elacestrant arm and 20% on the SOC arm developed grade 3 or 4 AEs. The most common AEs with elacestrant were GI-related, including nausea, vomiting, diarrhea, decreased appetite, and arthralgias. In addition, the duration of disease control with a CDK4/6 inhibitor was predictive of the benefit of elacestrant compared to SOC. Patients with at least 6 months, 12 months, and 18 months of prior CDK4/6 inhibitor use had median PFS of 2.79 vs. 1.91 months, 3.78 vs. 1.91 months, and 5.45 vs. 3.29 months, respectively, favoring elacestrant in each group [34]. Based on the results from the EMERALD trial, elacestrant was approved by the FDA for treating ER+ HER2− advanced breast cancer with the *ESR1* mutation and prior treatment with at least one line of ET.

There are a few ongoing studies with elacestrant. The ELEVATE study is a phase 1b/2, open-label, umbrella study where RP2D of elacestrant, combined with other drugs (alpelisib, everolimus, ribociclib, and palbociclib), will be identified and safety and efficacy assessed (NCT05563220) [35]. NCT04791384 is an ongoing phase Ib/II study investigating the combination therapy of elacestrant with abemaciclib in HR+ HER2− metastatic breast cancer with metastasis to the brain [36].

While early trials of second-generation SERDs were promising; the clinical benefits have been inconsistent in pivotal monotherapy studies, except in elacestrant. Further investigation is needed to determine whether combination therapies and (neo)adjuvant endocrine agents effectively treat MBC [37]. Common side effects, particularly gastrointestinal issues like nausea and vomiting, must also be discussed as they can affect the patient’s management and adherence to the treatment plan [38]. 

## 4. Other Oral SERDs in Development

### 4.1. Camizestrant (AZD9833)

Camizestrant is a pure ER antagonist and next-generational oral SERD. SERENA 1 is a phase I, open-label, dose-dependent exposure trial that analyzed the safety, tolerability, and preliminary clinical efficacy of camizestrant monotherapy (part A (escalation)/part B (expansion)) and in combination with palbociclib (parts C/D), everolimus (E/F), abemaciclib (G/H), and capivasertib (I/J) in pretreated women with ER+ HER2− advanced breast cancer. In parts A/B, treatment-related AEs were visual disturbances, bradycardia, nausea, fatigue, dizziness, and vomiting. Three doses, 75 mg QD, 150 mg QD, and 300 mg QD, were proposed for a phase II study. One out of 7, 2 out of 11, and 2 out of 10 patients achieved ORR with these doses, respectively, in this study [39]. SERENA-1 parts C/D is an ongoing study with a multi-part open-label trial currently examining camizestrant combined with palbociclib. Data from camizestrant 75 mg QD in combination with the standard dose of palbociclib have been presented [40]. Some patients experienced bradycardia, GI disturbances, and visual disturbances likely related to camizestrant; however, most were grade 1, and no patients required dose interruption or reduction for AEs in this study. The clinical benefit rate at 24 weeks for these heavily pretreated patients was 28% with this combination [40]. This combination is further analyzed in ongoing phase III studies of SERENA-4 and SERENA-6 [40]. 

SERNA-2 is a randomized, open-label, multicenter phase II trial that analyzed the efficacy and safety of camizestrant with the various dosages of 75–300 mg administered as a monotherapy in women with ER+, HER2− previously treated advanced breast cancer in comparison with fulvestrant. The camizestrant 300 mg QD arm was discontinued after 20 patients were enrolled in that arm. Camizestrant significantly reduced the risk of disease progression by 42% at a 75 mg dose ((HR 0.58, 0.41–0.81); *p* = 0.0124; *p* = 0.0124; PFS of 7.2 vs. 3.7 months) and 33% at a 150 mg dose ((HR 0.67, 0.48–0.92); *p* = 0.0161; PFS of 7.7 vs. 3.7 months) compared to fulvestrant [41]. Among the *ESR1* mutant tumors, median PFS with camizestrant 75 and 150 mg QD and fulvestrant were 6.3, 9.2, and 2.2 months, respectively, numerically favoring a benefit with camizestrant compared to fulvestrant, but this was not statistically significant as this phase II study was not powered to demonstrate benefit in the *ESR1* mutant subset. Among the *ESR1* wild-type tumors, the efficacy seemed similar numerically: 7.2, 5.8, and 7.2 months, respectively. Twelve percent of patients receiving 75 mg QD of camizestrant developed grade 3 or higher AEs compared to 13% receiving fulvestrant and 21% receiving camizestrant 150 mg QD. The most common AEs were bradycardia, visual disturbances, and fatigue. 

Multiple ongoing trials are investigating camizestrant further. SERENA-3 is a randomized, open-label, parallel-group pre-surgical trial looking at different camizestrant 75–150 mg doses in postmenopausal and possible premenopausal women with ER+/HER2− primary breast cancer [42]. SERENA-4 is an ongoing randomized multicenter, double-blind phase III trial comparing the safety and efficacy of camizestrant plus palbociclib vs. anastrozole plus palbociclib in patients with ER+/HER2− previously untreated breast cancer [43]. 

SERENA-6 is a novel and exciting phase III clinical trial where patients with advanced ER+ HER2− breast cancer who are receiving an AI plus palbociclib or abemaciclib and have developed the *ESR1* mutation but without overt disease progression will be randomized for a continuation of current treatment vs. switching AI to camizestrant with the continuation of the same CDK4/6 inhibitor [44]. The primary endpoint is PFS. 

Oral SERDs have evolved into a significant advance in oncology when treating ER+ breast cancer. The agents in this class, exemplified by camizestrant (AZD9833), provide a means of overcoming the limitations associated with the first generation of SERDs, such as fulvestrant, which has a low bioavailability due to its intramuscular administration [45]. A phase III trial of oral SERDs has demonstrated efficacy in modulating ER-regulated gene expression and antiproliferation in *ESR1* wild-type and mutant cells, demonstrating efficacy in treating early and advanced ER+ breast cancer. This addresses a critical need since *ESR1* mutations are a major cause of endocrine therapy resistance. Oral SERDs are combined with CDK4/6 inhibitors and PI3K/AKT/mTOR-targeted therapy to enhance treatment efficacy and overcome resistance. However, despite these advances, there remain challenges associated with optimizing pharmacokinetics and understanding resistance mechanisms [46,47]. Future research is positioned to address these hurdles to improve patient outcomes in ER+ breast cancer treatment.

### 4.2. Giredestrant

Giredestrant is another nonsteroidal SERD designed to target ER+ breast cancer [48]. GO39932 is a 1a/1b multicenter, open-label, dose-escalation study investigating the safety profile and preliminary antitumor activity of giredestrant alone and giredestrant in combination with palbociclib. Participants have ER+ HER2− advanced or metastatic breast cancer [49]. 

In GO39932 cohort A, giredestrant as a single agent was well tolerated at all doses without any DLTs and with low-grade AEs (i.e., nausea, arthralgia, and fatigue), none of which required participants to discontinue treatment in phase 1a. It had encouraging antitumor effects and clinical benefits at all doses [50]. In GO39932 cohort B, a giredestrant with palbociclib was assessed. None of the participants had to discontinue treatment due to AEs. Adverse effects in >/= 10% of patients included neutropenia, fatigue, bradycardia, diarrhea/constipation, dizziness/nausea, and so forth; 57% of patients had >/= grade 3 AEs, and 50% of patients had >/= grade 3 neutropenia. No drug–drug interactions were observed between giredestrant and palbociclib [48,50]. The clinical activity reported for the 30 mg monotherapy arm and the 100 mg dose with the the palbociclib arm was encouraging, with ORR at 20.0% and 47.7%, respectively. The clinical benefit rates were 55.0% and 81.3%, respectively [51]. 

Next, acelERA BC is a randomized, open-label, multicenter phase 2 study. It investigated the safety and efficacy of giredestrant compared to the physicians’ choice, ET (fulvestrant or an AI), for patients with ER+ HER2− locally advanced or metastatic breast cancer who have already been treated with one to two types of systemic therapy [52]. The trial could not meet the primary endpoint of investigator-assessed PFS; nevertheless, the giredestrant did show numerical improvement compared to ET in ORR and clinical benefit rate. Additionally, in patients with the *ESR1* mutation, their PFS was numerically higher with the giredestrant. Lastly, it was well tolerated and had a good safety profile consistent with others [53]. 

CoopERA is a completed randomized, open-label, multicenter phase 2 study investigating the efficacy, safety, and pharmacokinetics of giredestrant versus anastrozole within a 14-day window-of-opportunity phase followed by 16 weeks of a neoadjuvant treatment phase of giredestrant plus palbociclib vs. anastrozole plus palbociclib. The patient population is postmenopausal with ER+ HER2− untreated early breast cancer [54]. Giredestrant did indeed meet the primary endpoint and had higher Ki-67 suppression after week two compared to anastrozole. Note that this higher suppression continued up to surgery with giredestrant with palbociclib (−81% (95% CI: −86%, −75%)) versus anastrozole with palbociclib (−74% (95% CI: −80%, −67%)). Lastly, the safety profile is comparable to other giredestrant studies. This study shows that giredestrant demonstrates more antiproliferation in ER+ HER- early breast cancer than an aromatase inhibitor, anastrozole. Giredestrant with palbociclib also had a higher complete cell-cycle arrest at surgery, defined as Ki67 ≤ 2.7%, compared to anastrozole with palbociclib (20% vs. 14%) [55]. There are multiple ongoing studies investigating giredestrant in ER+ breast cancer where results are pending. 

Also, evERA is an ongoing randomized, open-label, multicenter phase III study that investigates the efficacy and safety of giredestrant with everolimus versus exemestane with everolimus. Participants are patients with ER+ HER2− locally advanced or metastatic breast cancer [56].

Finally, lidERA is an ongoing randomized, open-label, multicenter phase III study investigating the efficacy and safety of giredestrant versus the physician’s choice, endocrine therapy. Participants are patients with medium-risk and high-risk histologically proven Stage I to Stage III confirmed ER+ HER2− early breast cancer [56,57]. Both pre- and postmenopausal women are eligible for this study.

### 4.3. Amcenestrant

Amcenestrant is an orally bioavailable SERD with pure ER antagonism in vivo [58]. AMEERA-1 is an open-label, single-arm study that evaluated amcenestrant monotherapy in postmenopausal women with ER+/HER2− advanced breast cancer who were heavily pretreated. In AMEERA-1 arm 1 part A-B, the optimal dosing of 400 mg QB of amcenestrant monotherapy was chosen for a phase 2 dose with no grade ≥3 TRAEs like cardiac/eye toxicities reported. The confirmed objective response rate was 5/46 (10.9%), with an overall clinical benefit rate (CBR) of 13/46 (28.3%). The *wild-type ESR1* and mutated *ESR1* showed similar CBRs of 34.6% and 21.1%, respectively [59]. The result showed that amcenestrant at RP2D of 400 mg QD monotherapy demonstrated antitumor activity regardless of baseline *ESR1* mutation status and was well tolerated [59]. 

AMEERA-3 was a prospective, open-label, randomized phase 2 study to assess the safety and efficacy of amcenestrant compared to the ET of the physician’s choice in patients with ER+/HER2− metastatic or locally advanced breast cancer or metastatic breast cancer that progressed on ET. However, the trial did not meet its primary endpoint. The PFS was similar at 3.6 months for amcenestrant and 3.7 months for endocrine monotherapy [60]. TRAEs mainly were grade 1 or 2. In the I-SPY2 endocrine optimization protocol (EOP), the safety and efficacy of amcenestrant were evaluated with and without abemaciclib or letrozole [61]. The primary objective of the EOP is to assess the feasibility of treating molecularly selected patients with early stage ER+ HER2− molecular low-risk breast cancer with neoadjuvant endocrine therapy. 

AMEERA-4 (NCT04191382) was a phase II preoperative window of opportunity that compared the safety and efficacy of two dose levels of amcenestrant and a standard dose of letrozole with paired biopsies assessed for biomarkers in a 1:1:1 randomization design among early stage ER+ HER2− breast cancer patients [62]. The primary endpoint was the change from baseline in Ki67 after two weeks of treatment with amcenestrant or letrozole using paired biopsies. The reduction in Ki67 was 75.9% (67.9, 81.9) for amcenestrant 400 mg, 68.2% (58.4, 75.7) for amcenestrant 200 mg, and 77.7% (70.0, 83.4) for letrozole. All TRAEs were grade 1 or 2 and were similar among three arms, ranging from 20 to 25%. The sponsors prematurely discontinued this trial due to their strategic decision to stop the development of this drug. 

AMEERA-5, a phase III, randomized, double-blind, multinational study that analyzed women with ER+ HER2− metastatic breast cancer and compared amcenestrant plus palbociclib versus letrozole plus palbociclib failed to reach the prespecified boundary of continuation on interim analysis [63]. Based on this, the sponsor discontinued the global clinical development of amcenestrant.

### 4.4. Imlunestrant

Imlunestrant is another oral SERD in development. EMBER is an ongoing phase 1a/1b study with imlunestrant (doses 200–1200 mg) used as a monotherapy and combined with abemaciclib +/− an aromatase inhibitor (anastrozole, exemestane, or letrozole). In phase 1a, patients with ER+ HER2− advanced breast cancer with three or fewer prior therapies were recruited. Patients with ER+ endometrial cancer with prior platinum therapy were also recruited [64]. So far, monotherapy demonstrates no dose-dependent toxicities and TEAEs of mostly grade 1 or 2, including nausea, fatigue, and diarrhea. There was one grade 3 TEAE, diarrhea. Additionally, combination therapy (imlunestrant with abemaciclib +/− an aromatase inhibitor) in phase 1b has shown a satisfactory safety profile [65]. New findings presented at the 2022 San Antonio Breast Cancer Symposium regarding the combination therapy show favorable efficacy with a 12-month PFS compared to historical data from MONARCH 2 and 3 [66]. Another monotherapy trial is EMBER-2, an ongoing phase 1 study preoperative window studying the effects of imlunestrant on Stage I–III ER+ HER2− breast cancer in postmenopausal patients. As of yet, no results have been released [67]. 

EMBER-3 is an open-label, randomized 3-arm phase 3 study that compared the safety and efficacy of imlunestrant monotherapy vs. SOC ET (exemestane or fulvestrant) vs. imlunestrant plus abemaciclib among advanced ER+ HER2− breast cancer patients who have previously received an ET for advanced breast cancer [68]. EMBER-4 is another ongoing phase 3 study investigating imlunestrant compared to SOC ET in participants with high-risk ER+ HER2− early breast cancer. These participants will have already had two to five years of adjuvant endocrine therapy prior to this [69]. 

### 4.5. Rintodestrant

Rintodestrant is an oral SERD that competitively binds and degrades the estrogen receptor (ER). The safety and efficacy of rintodestrant were investigated in a phase 1 study, NCT03455270, among ER+ HER2− advanced breast cancer patients. Based on part 1 (dose escalation) and part 2 (dose expansion), the optimal dose of rintodestrant was 800 mg daily. Overall, 63% of patients developed TRAEs, and the most common TRAEs were hot flushes, fatigue, and nausea [70]. Part 3 of the study assessed the safety and efficacy of the combination of rintodestrant with palbociclib in patients with HR+/HER2− metastatic breast cancer that had already progressed on previous endocrine therapy. The overall CBR with this combination was 61% (61% among *ESR1* wild type, 56% *ESR1* mutant) [71]. 

### 4.6. AZD9496

AZD9496 is a preclinical compound that can be taken orally and is nonsteroidal, potent, and selective in the degradation and inhibition of ER activity [72]. Investigations of the effects of AZD9496, tamoxifen, and fulvestrant on estrogen-responsive genes in tumor samples were reported in a study. A human transcriptome array was used to measure mRNA levels in tumors treated with the drugs. Gene expression analysis was performed on tumors treated with the drugs. According to the study results, AZD9496 inhibits the expression of estrogen-responsive genes similarly to fulvestrant and tamoxifen. A dose-dependent inhibitory effect was also observed between fulvestrant and AZD9496 in MCF-7 cells compared to tamoxifen when estrogen-regulated genes were examined. A significant decrease in *ESR1* mRNA levels was not observed in vitro, suggesting that the protein downregulation previously described is due to a decrease in protein levels rather than an increase in transcript levels [73]. 

A preclinical study showed that AZD9496 could antagonize and degrade the estrogen receptor in breast cancer cell lines, xenograft models, and patient-derived xenografts with mutations in the *ESR1* gene. AZD9496 was more effective at inhibiting tumor growth when combined with the PI3K pathway inhibitors and a CDK4/6 inhibitor. Clinical trials of AZD9496 in phase I showed good tolerability and safety, and some patients who had been heavily pretreated could stabilize their disease for a prolonged period [73].

As part of a clinical trial conducted by Robertson et al. (NCT03236974), AZD9496 was compared with fulvestrant concerning the effects on changes in ER, progesterone receptors (PR), and Ki-67 biomarkers in patients who were newly diagnosed with ER+ HER2− breast cancer. A random allocation of patients was conducted between days 5 and 14 for treatment with AZD9496 and on day 1 for fulvestrant. Based on the study’s results, AZD9496 reduced the expression of ER, PR, and Ki-67, but the reductions were not superior to the fulvestrant ones. The plasma concentration of AZD9496 was lower than predicted, whereas the plasma concentration of fulvestrant was consistent with historical data. Neither treatment had significant safety concerns, and both were well tolerated [72].

## 5. Other SERDs in the Early Stage of Development

### 5.1. Borestrant

ZB716, also called borestrant, is a modified form of SERD fulvestrant with a boronic acid insertion, allowing it to be orally bioavailable. Borestrant has shown superiority compared to fulvestrant with better ER antagonism in preclinical data. ENZENO is an ongoing open-label, multicenter phase I/II study investigating borestrant’s properties as a monotherapy: safety, pharmacokinetics, pharmacodynamics, tolerability, and preliminary antitumor properties. Additionally, these properties will be investigated with combination therapy: borestrant with palbociclib. Participants are patients with ER+ HER2− locally advanced or metastatic breast cancer. Currently, no results have yet been released [74]. 

### 5.2. D-0502 (Taragarestrant)

This study, with identifier NCT03471663, is an ongoing open-label phase 1 study investigating the safety, pharmacokinetics, preliminary antitumor properties, and tolerability of D-0502 as a monotherapy, as well as in conjunction with palbociclib as a combination therapy. Participants are patients with ER+ HER2− advanced or metastatic breast cancer [75]. So far, results have shown good tolerability and preliminary clinical activity with D-0502. Regarding 22 patients who received monotherapy, preliminary efficacy results show an ORR of 5% and a CBR of 36%. Among 13 patients who received D-0502 in combination with palbociclib, preliminary results showed an ORR of 15% and a CBR of 77%. Additionally, there were no dose-limiting toxicities observed [76].

### 5.3. ZN-c5

ZN-c5 is another oral small-molecule antagonist and degrader of estrogen receptors that is being developed [77]. ZN-c5 has shown some activity in estrogen-dependent tumor models and has been well tolerated in clinical trials. In a clinical trial (NCT04514159), ZN-c5 was combined with abemaciclib. Patients may have undergone one previous hormonal-based therapy; however, they have not undergone any prior chemotherapy or received CDK4/6 inhibitors. The study was conducted in continuous 28-day cycles until disease progression or the occurrence of unacceptable toxicity. In the first clinical trial, ZN-c5 demonstrated good tolerance when administered at a dose of 50 mg [78].

Zn-c5 is an effective antagonist of the estrogen receptor in vivo and in vitro, as well as a potent degradative agent, and it is highly bioavailable compared to other SERDs. Oral administration of ZN-c5 at 5 mg/kg and 10 mg/kg resulted in significant inhibition of tumor growth in MCF-7 orthotopic tumor xenograft models, and ZN-c5 combined with CDK4/6 or PI3K inhibitors enhanced antitumor activity. The antitumor activity of ZN-c5 was also improved over that of fulvestrant in ER mutant models [77]. As a single agent and in combination studies, zinc-c5 is currently undergoing clinical trials, and its potency and degradation properties may potentially benefit patients with estrogen receptor-positive breast cancer (NCT03560531, NCT04514159) [78,79].

In addition to testing ZN-c5 on MCF-7 cells, its effects were tested on patient-derived xenograft models with ER mutations, such as WHIM20, a patient-derived model with the Y537S mutation in *ESR1*. ZN-c5 at 40 mg/kg reduced tumor growth by 64%, while fulvestrant at 200 mg/kg only reduced tumor growth by 13% (eight times higher than the clinical dose). According to these results, ZN-c5 exhibits superior antitumor properties compared with fulvestrant in xenograft models of human tumors [77].

In a clinical trial (NCT04514159), ZN-c5 was administered with abemaciclib. Patients may have undergone one previous hormonal-based therapy; however, they have not undergone any prior chemotherapy or received CDK4/6 inhibitors. The study was conducted in continuous 28-day cycles until disease progression or the occurrence of unacceptable toxicity. In the first clinical trial, ZN-c5 demonstrated good tolerance when administered at 50 mg [78]. This treatment demonstrates encouraging antitumor properties as an orally bioavailable SERD, thereby holding the potential to offer significant benefits to individuals afflicted with estrogen receptor-positive breast cancer.

It is generally accepted that patients prefer oral medications due to their ease of use and reduced frequency of hospital visits, both of which can significantly affect patients’ quality of life. Nevertheless, further detailed research and patient-reported outcome studies would help us understand the full scope of patient experiences and preferences, particularly compared to other treatment options [45].

## 6. Discussion

SERDs are an important endocrine therapy used to treat ER-positive breast cancer. Parenteral SERD fulvestrant has been approved and used in the treatment of metastatic ER-positive breast cancer for the last 2 decades. However, the need for large-volume intramuscular injections and poor bioavailability are important limitations of fulvestrant. In addition, in the current era of CDK4/6 inhibitors, the activity of fulvestrant on ER-positive breast cancer (both the wild-type *ESR1* and the *ESR1* mutant) after tumor progression on CDK4/6 inhibitors appears limited. 

The current trend of robust clinical studies for the development of oral SERDs may replace fulvestrant and improve the outcomes of patients with ER-positive metastatic breast cancer (Table 1). Elacestrant, a novel oral SERD, has already received approval from the FDA for the treatment of *ESR1* mutant ER-positive HER2-negative metastatic breast cancer after at least one line of endocrine therapy. Based on the elacestrant data, *ESR1* mutation and longer disease control with a CDK4/6 inhibitor appear to be predictive biomarkers for this oral SERD.

There are multiple ongoing clinical trials investigating the efficacy and safety of novel oral SERDS in the early stage and metastatic settings of ER+ breast cancer. Similarly, investigations are ongoing into using oral SERDs in combination with various targeted therapies. These studies will help shape the treatment landscape of ER+ breast cancer in the near future. For instance, EMBER-4 is a phase III clinical trial investigating the efficacy of imlunestrant compared to standard hormone therapy in patients with a history of high-risk ER+ HER2− breast cancer who have received standard endocrine therapy for two to five years (NCT05514054). The CAMBRIA 1 trial is similar to the EMBER-4 trial and compares camizestrant with standard endocrine therapy in adjuvant settings after patients have received at least 2 years of endocrine therapy (NCT05774951). 

Interestingly, the lidERA breast cancer trial and CAMBRIA 2 trials are phase III randomized trials of adjuvant giredestrant (lidERA) or camizestrant (CAMBRIA 2) vs. the adjuvant standard endocrine therapy of the physician’s choice among patients with a history of medium- to high-risk ER+ HER2− breast cancer (NCT04961996, NCT05952557). The primary endpoint is invasive-disease-free survival. The above studies, if they yield positive results, could expand the use of oral SERDs in adjuvant settings for the treatment of ER+ breast cancer in the near future. In the metastatic settings, various oral SERDs are being investigated in various lines of therapies and in combination with various cancer therapy partners, creating significant heterogeneity, as well as overlap. For instance, giredestrant in combination with palbociclib is being compared with letrozole plus palbociclib in the treatment of metastatic ER+ HER2− breast cancer in the first-line settings (persevERA Breast Cancer, NCT04546009). SERENA -4 and EMBER 3 trials (NCT04711252, NCT04975308 ) are similar to the persevERA trial design except that they investigate camizestrant plus palbociclib (SERENA 4) and imlunestrant plus abemaciclib (EMBER 3). 

Similarly, the pionERA breast cancer trial (NCT06065748) compares giredestrant plus a CDK4/6 inhibitor of physicians’ choice with fulvestrant plus a CDK4/6 inhibitor of physicians’ choice in the first-line treatment of ER+ HER2− metastatic breast cancer. However, only patients with tumors resistant to prior endocrine therapy and with confirmed *ESR1* mutations are allowed. Additionally, the evERA breast cancer trial (NCT05306340) compares giredestrant plus everolimus with the physicians’ choice, endocrine therapy plus everolimus, in later than first-line settings of ER+ HER2− metastatic breast cancer. 

Fulvestrant still has a role in treating ER-positive HER2-negative metastatic breast cancer, especially those with wild-type *ESR1*, often in combination with targeted therapy. In addition, clinicians may choose fulvestrant in clinical scenarios where compliance with oral medication is questionable. Similarly, elacestrants (and future oral SERDs, if approved) are prescription drugs, while fulvestrants are generally administered in oncology clinics. Insurance coverage and out-of-pocket cost variations between oral SERDs and fulvestrant may exist. This should also be explored and discussed with the patients. 

From a safety standpoint, adverse effects associated with oral SERDs seem to be manageable. As the results of ongoing studies with novel oral SERDs (as a single agent or in combination with other targeted therapies), the landscape of endocrine-therapy-based treatment for ER-positive HER2-negative breast cancer may continue to evolve in the next few years. There is a lack of data on the activity of oral SERDs in *PIK3CA* mutant ER-positive HER2-negative metastatic breast cancer. Similarly, how already-approved antibody–drug conjugates and those in development will fit into the sequence of treatments with oral SERDs is unknown at this time [80,81]. Current ongoing studies and exploratory translational studies may help us understand these areas.

## 7. Conclusions

In this paper, we have discussed current preclinical and clinical data on novel oral SERDs. Many oral SERDs demonstrate meaningful efficacy and manageable safety in early stage clinical trials. As of May 2023, elacestrant remains the only oral SERD to have received FDA approval in the treatment of ER-positive HER2-negative metastatic breast cancer. Results of ongoing studies may lead to the approval of more oral SERDs as a single agent or as a part of combination therapy in this space. 

## Figures and Tables

**Figure 1 cancers-16-00619-f001:**
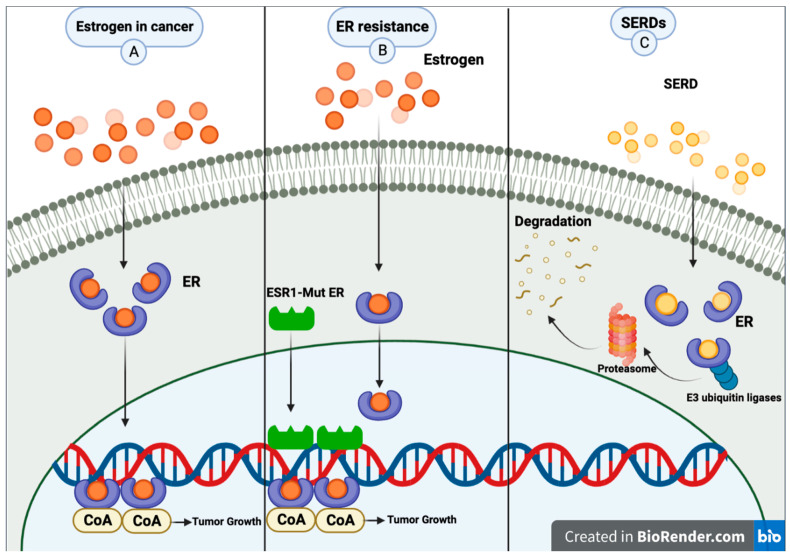
(**A**) Common pathway for estrogen in breast cancer. (**B**) Metastatic breast cancer patients may experience resistance mechanisms to endocrine therapy. A mutation of estrogen receptor 1 (*ESR1*) causes constant ER activity and enhanced transcription of ER-dependent genes without hormones, resulting in resistance to estrogen deprivation and aromatase inhibitor therapy. (**C**) SERDs bind to the estrogen receptor; then, E3 ubiquitin ligases and ubiquitinates the ER, marking it for degradation by the proteasome. The proteasome eventually degrades the ubiquitinated ER. Created with BioRender.com (accessed on 8 June 2023).

**Table 1 cancers-16-00619-t001:** Summary of the clinical trial of oral SERDs with efficacy and safety features.

Trial (Identifier)	Status	Study Design	Study Drug	Number Enrolled	Disease Setting	Study Population	Results	Safety
RAD1901-005 (NCT02338349)	Completed	Phase I Single group, open-label	Elascetrant (400 mg daily)	57	LABC or mBC	Postmenopausal patients with ER+ HER2− BC	ORR 19.4% in patients with RP2D, 15% in patients with prior SERD use, 16.7% in patients with prior CDK4/6i use, and 33.3% in patients with *ESR1* mutation. The clinical benefit rate was 42.6% overall.	No DLTs, most common AEs include nausea (33%), increased blood triglycerides (25%) and decreased blood phosphorous (25%); most AEs were grade 1–2.
RAD1901-106 (NCT02650817)	Completed	Phase Ib Two-cohort, open-label	Elascetrant (200–400 mg daily)	16	LABC or mBC	Postmenopausal patients with ER+ HER2− BC	Median reduction in tumor FES uptake was 89.1%. ORR was 11.1% and clinical benefit rate was 30.8%	Common adverse effects include nausea (68.8%), fatigue (50%), dyspepsia (43.8%). Most AEs were grade 2 in severity.
SOLTI-1905 ELIPSE (NCT04797728)	Completed	Phase I Single group, open-label	Elascetrant (400 mg daily for 4 weeks)	23	Early resectable breast cancer	Treatment naive postmenopausal patients with ER+ HER2− BC	After 4 weeks, CCCA was achieved in 27% of patients. A 41% relative reduction in Ki-67 from baseline was observed	1 TRAE event requiring the patient to discontinue treatment. Most AEs were hot flush (*n* = 6), hot flush (*n* = 6), and constipation (*n* = 2), all of which were grade 1.
NCT04791384	Recruiting	Phase Ib/II Single group, open-label	Elascetrant + Abemaciclib	44	mBC	ER+ HER2− BC with brain metastases and ≤2 prior lines of treatment	N/A	N/A
EMERALD (NCT03778931)	Active, not recruiting	Phase III Randomized, open-label	Elascetrant vs. SOC (i.e., fulvestrant, anastrozole, letrozole, exemestane)	478	LABC or mBC	Postmenopausal patients with ER+ HER2− BC	The longer the prior CDK4/6i duration, the increased PFS observed with elacestrant vs. SOC. With a duration on CDK4/6i of at least 12 months, the median PFS was 8.6 months with elacestrant compared to 2.1 months with SOC.	Most AEs were grade 1–2, such as nausea. There were no grade 4 TRAEs reported. 3.4% of patients with elacestrant had to discontinue therapy versus 0.9% of patients with SOC. No deaths were observed in either arm.
GO39932 Cohort A (NCT03332797)	Active, not recruiting	Phase I Non-randomized, open-label	Giredestrant (10–250 mg)	181	LABC or mBC	ER+ HER2− BC	ORR at 20.0% and 47.7% respectively. Clinical benefit rates were 55.0% and 81.3% respectively. Of the FES-PET avid disease patients, 78% showed complete or near complete suppression of FES uptake.	There are no DLTs and low-grade AEs (i.e., nausea, arthralgia, fatigue). None of the AEs required treatment termination.
acelERA BC (NCT04576455)	Active, not recruiting							
GO39932 Cohort B (NCT03332797)								
MORPHEUS-BREAST CANCER (NCT04802759)	Recruiting	Phase Ib/II Randomized, open-label, umbrella study	Giredestrant (30 mg) +/− abemaciclib/ipatasertib/inavolisib/ribociclib/everolimus/samuraciclib/PH FDC SC/PH FDC SC + abemaciclib/PH FDC SC + palbociclib	510	Inoperable LABC or mBC	Cohort 1: ER+ HER2− BC who have shown disease progression during or after CDK4/6 inhibitor treatment in either 1st- or 2nd-line setting. Cohort 2: ER+ HER2+ BC who have shown disease progression during or after SOC anti-HER2 therapy (i.e., trastuzumab-and-taxane-based therapy), HER2−targeting ADC (i.e., ado-trastuzumab emtansine or trastuzumab-deruxtecan), or HER2−targeting TKIs (i.e., tucatinib, lapatinib, pyrotinib, or neratinib)	N/A	N/A
CoopERA (NCT04436744)								
evERA (NCT05306340)	Recruiting	Phase III, Randomized, open-label	Giredestrant (30 mg) + everolimus (10 mg) for 4 weeks versus Exmestane (25 mg) + everolimus (10 mg) for 4 weeks	320	LABC or mBC	ER+ HER2− BC participants with prior treatment of CDK4/6 inhibitors and endocrine therapy	N/A	N/A
heredERA (NCT05296798)	Recruiting	Phase III Randomized, open-label	Giredestrant + Phesgo versus ET (investigator’s choice) + Phesgo after induction therapy with a taxane + Phesgo	812	LABC or mBC	ER+ HER2+ BC that is previously untreated	N/A	N/A
persevERA (NCT04546009)	Recruiting	Phase III, Randomized, double-blind	Giredestrant (30 mg daily) + palbociclib (125 mg for 3 weeks in every 4 week cycle) versus letrozole (2.5 mg daily) + palbociclib (125 mg for 3 weeks in every 4 week cycle)	978	LABC or mBC	ER+ HER2− recurrent or progressed BC	N/A	N/A
lidERA (NCT04961996)	Recruiting	Phase III Randomized, open-label	Giredestrant (30 mg daily) versus physician’s choice ET for 5 years (if tolerated)	4100	early BC	Medium- to high-risk Stage I-III histologically confirmed ER+ HER2− BC	N/A	N/A
SERENA-1 part A (NCT03616587)	Ongoing	Phase I FIH 1, Open Label	camizestrant (25–450 mg daily)	403	LABC or mBC	Women with HR+/HER2− BC	85% loss of m*ESR1*,ORR 16.3%, CBR 42.3% (50% *mESR1*), mPFS 5.5 mo	DLT at 300 mg and 450 mg. G1: Visual disturbances, bradycardia, nausea, fatigue, dizziness, vomiting, asthenia.
SERENA-2 (NCT04214288)	Active, not recruiting	Phase II randomized, open-label	camizestrant (75–300 mg) vs. fulvestrant	240	LABC or mBC	Postmenopausal women with HR+/HER2− BC	Camizestrant 75 mg reduced the risk of disease progression by 42% at a 75 mg dose ([HR] 0.58, *p* = 0.0124; median PFS of 7.2 versus 3.7 months 33% at a 150 mg dose HR 0.67, PFS of 7.7 versus 3.7 months	TRAE’s ≥ 3 75ng: BP increase (1.3%), fatigue, nausea, anemia, arthralgia, alanine transaminase increase, extremity pain, hyponatremia, 150 mg BP increase of 1.3%, 300 mg diarrhea (5.0%) and BP increase (5.0%), fulvestrant-anemia (2.7%).
SERENA-3	Recruiting	Phase II randomized, open-label	camizestrant (75–150 mg) with 5 day washout before surgery	132	LABC or mBC	Postmenopausal (possibly premenopausal) women with HR+/HER2− BC	N/A	N/A
SERENA-1 parts C/D (NCT03616587)	Ongoing	Phase I Multiple parts, open label	camizestrant + palbociclib	305 (75 mg parts C/D 25)	LABC or mBC	Women with ER+/HER2− BC	ORR 12%, 24 weeks CBR 28%	DLT at 150 mg dose. 75 mg cohorts G ≥ 3: neutropenia (68%). 75 mg cohorts all G: neutropenia (80%), visual disturbances (44%), fatigue (20%), infections (20%), anemia (20%), bradycardia (16%), nausea (16%), decreased appetite (12%), diarrhea (12%), vomiting (12%).
SERENA-4 (NCT04711252)	Recruiting	Phase III randomized, double blind	camizestrant (75 mg daily) + palbociclib (125 mg 3w/1w) vs. anastrozole (1 mg daily) + palbociclib Men and premenopausal patients receive LHRH agonist in addition to study treatment	1342	LABC or mBC	De novo or recurrent ER+/HER2− BC.	N/A	N/A
SERENA-6 (NCT04964934)	Ongoing	Phase III randomized, double-blind	camizestrant + CDK4/6i (palbociclib or abemaciclib) vs. ongoing treatment with AI (anastrozole or letrozole) + CDK4/6	300	LABC or mBC	ER+/HER2− BC is on the current 1 L SOC. Detectable *ESR1* mutation.	N/A	N/A
NCT03455270 part 3	Active recruitment completed	Phase I Single Group, Open Label	rintodestrant (800 mg daily) + palbociclib (125 mg 3w/1w)	107	LABC or mBC	Women with ER+/HER2− BC	Palbociclib w/rintodestrant CBR of 60% compared to rintodestrant alone of 28%. ORR 5% (4% WT *ESR1*, 6% m*ESR1*), CBR 60% (61% WT *ESR1*, 56% m*ESR1*), mPFS 7.4 mo	G ≥ 3: neutropenia (53%), leukopenia (18%). All G: neutropenia (90%), leukopenia (45%), anemia (15%), thrombocytopenia (10%), asymptomatic bacteriuria (10%).
AMEERA-1 arm 1 part A-B	Active, not recruiting	Phase I/II Randomized, Open Label	amcenestrant (200–600 mg daily)	136	LABC or mBC	Women with ER+/HER2− BC	ORR 5/46 10.9%, (CBR) of 13/46 (28.3%). *ESR1* wild type and mutated *ESR1* showed similar CBR (34.6% and 21.1% respectively	No Grade ≥ 3 TRAEs or clinically significant cardiac/eye toxicities were reported.
AMEERA-3 (NCT04059484)	Active, not recruiting	Phase II Randomized, Open Label	AMC 400 mg WD or Single agent TPC (fulvestrant, aromatase inhibitor, or tamoxifen	367	LABC or mBC	Postmenopausal women, premenopausal, or men taking GnRH agonists with ER+/HER2− aBC who received <2 prior ET and ≤1 prior chemotherapy or ≤1 targeted therapy for aBC.	Study did not meet the primary objective; PFS per ICR was numerically similar between AMC and TPC-PFTS 3.6 vs. 3.7 months, HR 1.051	0.7% Grade ≥ 3 TRAEs; TRAE’s with AMV vs. TPC Grade 1/2: Nausea (14.0% vs. 4.1%), vomiting (8.4% vs. 1.4%), hot flush (8.4% vs. 7.5%), asthenia (7.0% vs. 1.4%), fatigue (5.6% vs. 6.1%), and injection site pain (0% vs. 6.8%); 4.9%.
AMEERA-1 arm 2–5								
I-SPY1 EOP (NCT01042379)	Recruiting	Phase II Randomized, open label	amcenestrant (200 mg daily) ± abemaciclib/letrozole for 6 mo	5000	early stage BC neoadjuvant	Clinical high-risk and molecular low-risk (MammaPrint^®^ low-risk score) ER+/HER2− BC (≥2.5 cm)	N/A	N/A
AMEERA-5 (NCT04478266)	Active, not recruiting	Phase III Randomized, double blind	amcenestrant (200 mg daily) + palbociclib vs. letrozole + palbociclib	1068	LABC or mBC	ER+/HER2− BC. ECOG 0–2	Interim analysis showed negative results and an independent data-monitoring committee found that amcenestrant, combined with palbociclib, did not meet the criteria for continuation compared to the control arm.	No new safety signals were observed.
AMEERA-6 (NCT05128773)	Terminated	Phase III Randomized, double blind	amcenestrant vs. tamoxifen 5 years	3	early stage BC adjuvant	Stage IIB or III ER+/HER2± BC undergone surgery and adjuvant RT if indicated.	Study Terminated	Study Terminated
EMBER (NCT04188548)	Recruiting	Phase Ia/Ib Randomized, open label. multi-cohort	Imlunestrant (200–1200 mg daily) +/− everolimus, abemaciclib, alpelisib, trastuzumab	500	LABC or mBC and other select non-breast cancer	HR+ HER2− BC with 3 or fewer prior lines of treatment and ER+ endometrial cancer previously treated with platinum therapy	Combination therapy also shows positive antitumor properties with a 12-month PFS in comparison to MONARCH 2 and 3	Monotherapy demonstrates no dose-dependent toxicities; TEAEs mainly were grade 1–2 (i.e., nausea, fatigue, diarrhea). Combination therapy showed an adequate safety profile.
EMBER-2 (NCT04647487)	Active, not recruiting	Phase I Randomized, open-label	Imlunestrant	90	early stage (I–III) BC neoadjuvant	Post-menopausal women with ER+ HER2− BC	N/A	N/A
EMBER-3 (NCT04975308)	Recruiting	Phase III Randomized, open-label	Imlunestrant +/− abemaciclib versus ET (investigator’s choice)	860	LABC or mBC	ER+ HER2− previously treated with ET	N/A	N/A
EMBER-4 (NCT05514054)	Recruiting	Phase III Randomized, open-label	Imlunestrant versus ET (investigator’s choice)	6000	Early BC with increased risk of recurrence	Patients with ER+ HER2− BC who received 2–5 years of adjuvant ET	N/A	N/A
ENZENO (NCT04669587)	Recruiting	Phase I/II Non-randomized, open-label	Borestrant +/− palbociclib	106	LABC or mBC	ER+ HER2− BC	N/A	N/A
NCT03471663	Active, not recruiting	Phase I Randomized, open-label, multi-parts	D-0502 +/− palbociclib	200	LABC or mBC	ER+ HER2− BC	Radiological tumor response and CBR observed in both monotherapy and combination therapy	No DLTs
NCT04514159	Ongoing	Phase I/II Open-label, multicenter	ZN-c5	181	mBC	Postmenopausal patients, adenocarcinoma of the breast ER+ HER2− BC	N/A	N/A
CAMBRIA-1 (NCT05774951)	Recruiting	Phase III Open-label, multicenter	Arm A: Standard ET of choice Arm B: Camizestrant	4300	Early BC with intermediate-high risk or increased risk of recurrence	ER+/HER2− early breast cancer who received locoregional therapy and ET for 2–5 years.	N/A	N/A
CAMBRIA-2 (NCT05952557)	Recruiting	Phase III Open-label	Camizestrant vs. ET as adjuvant therapy	5500	Early BC with intermediate-high risk or increased risk of recurrence	ER+/HER2− early breast cancer who received locoregional therapy	N/A	N/A

N/A: Non applicable.

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
