# Peer review of "Oral SERD, a Novel Endocrine Therapy for Estrogen Receptor-Positive Breast Cancer"

_cancers, 2024, doi:10.3390/cancers16030619_

Round 1

Reviewer 1 Report

Comments and Suggestions for Authors

Fulvestrant is an ERa degrador synthetised some years ago prompting  to the hypothesis that it might be clinically  usefull  for the treatment of ERa-posisive breat cancers  resistant to conventional  endocrine treatments.The fact that this drug might solely be admistrtrated by intravenous injection led to search for derivatves  or analogs for oral medications. Present review focusses on this topic :  properties  of several compounds are extensively described .   Data suggest  the  an  usefull  fulvestrant substitute is still non  evident( personnal opinion) .   Accessibility to the pesent report  to clinicians seems to me  of importance. 

A point to be stressed, usually not evoked  in the medical literaure, is  the  established evidence that the  major antitumor activity of fulvestrant is not its ERa degradation  but its capacity to  trasiently immobilise  the receptor within the cell nucleus  implying SUMOylations with concomitant  transcriptional disturbances, a step preceeding the receptor elimination. Hence, to limit studies as  described in the present report  would largely be insufficient . Elaboationof future  programs must be adapted to present knowledge  . Note  in this context ( example) that a stuctural mofification of the side chain of fulvestrant  responsible of its   ERa degradation ability may eliminate is degradative   activity without strongly suppress its antitumor activty  ( (Agourida V; J Med Chem2009, 52? 883-887 )

 I strongly suggest to the authors of this publication, to evoke this   thematic developped by several proups: DP Mc Donnell ( Wandell SE Biochem Pharmacol 82:122, 2011) ;Guan J Cell178:949, 2019 ( important) ;  S Mader ( Traboulsi T  Oncogene 38:1019, 2019; Vallet A J Biol Chem229:102757, 2023); Wang  Q  Gland Surg 12: 963,2023        This listing may orient the research .  which must  not be exhaustive but informative to provide  valuable complementary  interest.

Author Response

Reviewer #1:

[Fulvestrant is an ERa degrador synthetised some years ago prompting  to the hypothesis that it might be clinically  usefull  for the treatment of ERa-posisive breat cancers  resistant to conventional  endocrine treatments.The fact that this drug might solely be admistrtrated by intravenous injection led to search for derivatves  or analogs for oral medications. Present review focusses on this topic :  properties  of several compounds are extensively described .   Data suggest  the  an  usefull  fulvestrant substitute is still non  evident( personnal opinion) .   Accessibility to the pesent report  to clinicians seems to me  of importance. 

A point to be stressed, usually not evoked  in the medical literaure, is  the  established evidence that the  major antitumor activity of fulvestrant is not its ERa degradation  but its capacity to  trasiently immobilise  the receptor within the cell nucleus  implying SUMOylations with concomitant  transcriptional disturbances, a step preceeding the receptor elimination. Hence, to limit studies as  described in the present report  would largely be insufficient . Elaboationof future  programs must be adapted to present knowledge  . Note  in this context ( example) that a stuctural mofification of the side chain of fulvestrant  responsible of its   ERa degradation ability may eliminate is degradative   activity without strongly suppress its antitumor activty  ( (Agourida V; J Med Chem2009, 52? 883-887 )

 I strongly suggest to the authors of this publication, to evoke this   thematic developped by several proups: DP Mc Donnell ( Wandell SE Biochem Pharmacol 82:122, 2011) ;Guan J Cell178:949, 2019 ( important) ;  S Mader ( Traboulsi T  Oncogene 38:1019, 2019; Vallet A J Biol Chem229:102757, 2023); Wang  Q  Gland Surg 12: 963,2023        This listing may orient the research .  which must  not be exhaustive but informative to provide  valuable complementary  interest.]

Response: Thank you for your insightful comments and suggestions regarding our manuscript. We appreciate the opportunity to enhance our work based on your expert feedback. Below, we address each of your points:

Furthermore, research indicates that fulvestrant-resistant breast cancer cells are capable of proliferating in a manner independent of estrogen signaling and can display downregulated estrogen signaling.(20, 21) This information is crucial to guiding clinical application and future research in this area, as it provides insights into the drug's mechanism of action and resistance patterns in breast cancer cells.

We acknowledge your recommendation to include the thematic development by various groups such as DP McDonnell, Guan, S Mader, and Wang Q. Your suggestion to reference specific studies (e.g., Agourida V; J Med Chem 2009, 52, 883-887) is well-taken. We revised our manuscript to include these references, ensuring that our review is informative and provides a comprehensive perspective on the current state of research in this area.

Studies show, however, that fulvestrant's primary anticancer effect is not through ER degradation, but rather through the transient immobilization of ER receptor in the nucleus.(22) A SUMOylation process takes place in this action, which leads to transcriptional disturbances, as well as the elimination of receptors.(23, 24, 25) There have been studies showing that fulvestrant induces transient binding of ERα to DNA, followed by rapid release without loss of nuclear localization. Unlike proteasome inhibition, this mechanism occurs prior to receptor degradation.(23) With fulvestrant, chromatin accessibility is reduced, suggesting chromatin remodeling at ERα target regions prevents transcription despite receptor binding. The SUMO2/3 marks have also been found on chromatin in cells treated with fulvestrant, but not with SERMs.(23, 26)

As a result of further study, structural modifications to the side chain of fulvestrant have been found to eliminate the degradation abilities of the agent, yet do not significantly affect its anticancer potential. In this manner, we can infer that fulvestrant has a complex interplay between the molecular actions it exerts against ER-positive breast cancer, including SUMOylation, that is responsible for its overall effectiveness in combating this disease.(22, 27)

Reviewer 2 Report

Comments and Suggestions for Authors

1.  It would be helpful to provide a more detailed description of the specific objectives and research questions that your review aims to address. Are you primarily focusing on the mechanism of action of oral SERDs, their clinical efficacy, or any specific challenges in their development?

2. While you mention ESR1 mutations as a common mechanism of endocrine therapy resistance, consider elaborating on the significance of these mutations, their prevalence, and how they impact treatment decisions. This will provide readers with a better understanding of the problem you're addressing.

3. Ensure that your review includes the most up-to-date research findings and developments related to oral SERDs. The field of breast cancer treatment is rapidly evolving, so it's crucial to incorporate recent studies and clinical trial results.

4. While you mention that second-generation SERDs have improved oral bioavailability and pharmacokinetics, provide a more comprehensive discussion of their limitations or potential side effects. This will give a balanced view of their use in clinical practice.

5. Consider including information on patient experiences and preferences regarding oral SERDs compared to other treatment options. This can add a patient-centered perspective to your review.

6.  Provide more insights into the potential impact of oral SERDs on the overall management of ER+ breast cancer. Discuss any ongoing research, challenges, or innovations that might shape the future of this field.

7. Ensure that the content flows smoothly and follows a clear structure. Use subheadings to separate different sections of your review, such as background, current status, and future perspectives, to make it easier for readers to navigate and understand the information presented.

Comments on the Quality of English Language

Moderate editing of the English language is required.

Author Response

Reviewer #2:

[1.  It would be helpful to provide a more detailed description of the specific objectives and research questions that your review aims to address. Are you primarily focusing on the mechanism of action of oral SERDs, their clinical efficacy, or any specific challenges in their development?]

Thank you for your insightful comments. Our review primarily focuses on three key areas concerning SERDs: their mechanism of action, clinical efficacy, and the challenges encountered in their development. Specifically, we aim to:

Mechanism of Action: Elaborate on how oral SERDs function at the molecular level to degrade estrogen receptors, overcoming resistance mechanisms that limit the efficacy of traditional endocrine therapies.

Clinical Efficacy: Provide a comprehensive overview of current clinical trials and studies, highlighting the efficacy of oral SERDs in treating estrogen receptor-positive breast cancer, especially in cases where resistance to conventional endocrine therapies has developed.

Development Challenges: Discuss the challenges faced during the development of oral SERDs, including issues related to pharmacokinetics, bioavailability, and patient compliance, and how these challenges are being addressed to optimize treatment outcomes.

Our review synthesizes recent advancements and ongoing research in this field, aiming to provide a thorough understanding of oral SERDs as a promising treatment option for estrogen receptor-positive breast cancer.

We added this paragraph in the end of the introduction:

The aim of our review is to provide a comprehensive analysis of SERDs in treating estrogen receptor-positive breast cancer. We focus on elucidating the molecular mechanisms of SERDs, assessing their clinical efficacy in various therapeutic contexts, and identifying the challenges in their development and implementation. This review focuses on to synthesize current research and advancements, offering insights into the potential of oral SERDs as an effective treatment option.

[2. While you mention ESR1 mutations as a common mechanism of endocrine therapy resistance, consider elaborating on the significance of these mutations, their prevalence, and how they impact treatment decisions. This will provide readers with a better understanding of the problem you're addressing.]

ESR1 mutations play a crucial role in development of endocrine therapy resistance in breast cancer. ESR mutations are expressed in the majority of breast cancers and are regulators of breast cancer development and progression. ESR1 mutations, particularly ones in the ligand binding domain have accounted for acquired endocrine resistance in a large fraction of patients with metastatic disease (19).

[3. Ensure that your review includes the most up-to-date research findings and developments related to oral SERDs. The field of breast cancer treatment is rapidly evolving, so it's crucial to incorporate recent studies and clinical trial results.]

We appreciate your valuable suggestion to include the latest research findings and developments related to oral SERDs in breast cancer treatment. In response to your comment, we have conducted an extensive update of our literature review, incorporating recent studies and clinical trial results. This includes the addition of recent FDA approvals, comparative analysis of new oral SERDs versus existing therapies, and a discussion on their efficacy and safety profiles. These updates ensure our manuscript reflects the current state of research and provides a comprehensive overview of the rapidly evolving field of oral SERDs in breast cancer treatment.

[4. While you mention that second-generation SERDs have improved oral bioavailability and pharmacokinetics, provide a more comprehensive discussion of their limitations or potential side effects. This will give a balanced view of their use in clinical practice.]

Thank you for your comment. We added following paragraph in our manuscript:

While early trials of second-generation SERDs were promising, the clinical benefits have been inconsistent in pivotal monotherapy studies, with the exception of elacestrant. It is clear that further investigation is needed in order to determine whether combination therapies and (neo)adjuvant endocrine agents are effective in treating MBC. (35) The occurrence of common side effects, particularly gastrointestinal issues like nausea and vomiting, must also be discussed as they can affect the management of the patient as well as the adherence to the treatment plan.(36)

[5. Consider including information on patient experiences and preferences regarding oral SERDs compared to other treatment options. This can add a patient-centered perspective to your review.]

Thank you for your valuable suggestion to include information on patient experiences and preferences regarding oral SERDs compared to other treatment options. We recognize the importance of incorporating a patient-centered perspective in our review. While our manuscript currently focuses on the clinical and scientific aspects of oral SERDs, we acknowledge that understanding patient perspectives is crucial in the context of treatment choices. In the manuscript we have highlighted the need for further studies on this area.

It is generally accepted that oral medications are preferred by patients due to their ease of use and reduced frequency of hospital visits, both of which can make a significant difference in patient quality of life. Nevertheless, further detailed research and patient-reported outcome studies would be helpful in understanding the full scope of patient experiences and preferences, particularly when compared to other treatment options.(75)

[6.  Provide more insights into the potential impact of oral SERDs on the overall management of ER+ breast cancer. Discuss any ongoing research, challenges, or innovations that might shape the future of this field.]

There are multiple clinical trials ongoing investigating the efficacy and safety of novel oral SERDS in the early stage and metastatic settings of ER+ breast cancer. Similarly, investigations are ongoing into using oral SERDs in combination with various targeted therapies. These studies will help shape the treatment landscape of ER+ breast cancer in the near future. For instance, EMBER-4 is a phase III clinical trial investigating the efficacy of imlunestrant compared to standard hormone therapy in patients with a history of high-risk ER+ HER2- breast cancer who have received standard endocrine therapy for two to five years (NCT05514054). CAMBRIA 1 trial is similar to the EMBER-4 trial and compares camizestrant with standard endocrine therapy in adjuvant settings after patients have received at least 2 years of endocrine therapy (NCT05774951).

Interestingly, the lidERA Breast Cancer trial and CAMBRIA 2 trials are phase III randomized trials of adjuvant giredestrant (lidERA) or camizestrant (CAMBRIA 2) vs. adjuvant standard endocrine therapy of physician’s choice among patients with a history of medium to high risk ER+ HER2- breast cancer (NCT04961996, NCT05952557). The primary endpoint is invasive disease-free survival. The above studies, if they yield positive results, could expand the use of oral SERDs in adjuvant settings for the treatment of ER+ breast cancer in the near future. In the metastatic settings, various oral SERDs are being investigated in various lines of therapies and in combination with various cancer therapy partners, creating significant heterogeneity as well as overlap. For instance, giredestrant in combination with palbociclib is being compared with letrozole plus palbociclib in the treatment of metastatic ER+ HER2- breast cancer in the first-line settings (persevERA Breast Cancer, NCT04546009). SERENA -4 and EMBER 3 trials (NCT04711252, NCT04975308 ) are similar to the persevERA trial design except that they investigate camizestrant plus palbociclib (SERENA 4) and imlunestrant plus abemaciclib (EMBER 3).

Similarly, the pionERA Breast Cancer trial (NCT06065748) compares giredestrant plus a CDK4/6 inhibitor of physicians’ choice with fulvestrant plus a CDK4/6 inhibitor of physicians’ choice in the 1st line treatment of ER+ HER2- metastatic breast cancer. However, only patients with tumors resistant to prior endocrine therapy and with confirmed ESR1 mutations are allowed. Additionally, evERA Breast Cancer trial (NCT05306340) compares giredestrant plus everolimus with physicians’ choice endocrine therapy plus everolimus in later than first-line settings of ER+ HER2- metastatic breast cancer.

[7. Ensure that the content flows smoothly and follows a clear structure. Use subheadings to separate different sections of your review, such as background, current status, and future perspectives, to make it easier for readers to navigate and understand the information presented.]

Thank you for your valuable feedback. We agree that a clear structure and smooth flow of content are crucial for readability and comprehension. In each heading we have this flow and we have tried to review the most information thorough various studies in this topic.

Reviewer 3 Report

Comments and Suggestions for Authors

The development of oral selective estrogen receptor degraders (SERDs) signifies a significant stride in tackling drug resistance within estrogen receptor-positive (ER+) breast cancer, especially linked to ESR1 mutations. While first-generation SERDs like fulvestrant have demonstrated efficacy against ESR1 mutant tumors, their utility is hampered by poor bioavailability and the requirement for intramuscular injection. The ongoing phase III trials of second-generation oral SERDs present a promising prospect, as they not only enhance convenience but also improve effectiveness, potentially offering an optimized therapeutic approach for patients dealing with both early and advanced ER+ breast cancer. Additionally, the authors should address a couple of minor comments.

Minor Comments:

1. In the introduction, the author should incorporate a mention of the role of estrogen receptor (ER) not only in the nucleus but also in the cytoplasm, where it may function as a post-transcriptional regulator.

2. The author should highlight the significance of selective estrogen receptor degraders (SERDs) in the context of heterodimerization, particularly addressing the impact of ER-alpha heterodimerization with other proteins. This would provide a more comprehensive understanding of the mechanisms underlying the therapeutic efficacy of SERDs.

Author Response

Reviewer #3:

[The development of oral selective estrogen receptor degraders (SERDs) signifies a significant stride in tackling drug resistance within estrogen receptor-positive (ER+) breast cancer, especially linked to ESR1 mutations. While first-generation SERDs like fulvestrant have demonstrated efficacy against ESR1 mutant tumors, their utility is hampered by poor bioavailability and the requirement for intramuscular injection. The ongoing phase III trials of second-generation oral SERDs present a promising prospect, as they not only enhance convenience but also improve effectiveness, potentially offering an optimized therapeutic approach for patients dealing with both early and advanced ER+ breast cancer. Additionally, the authors should address a couple of minor comments.

Minor Comments:

  1. In the introduction, the author should incorporate a mention of the role of estrogen receptor (ER) not only in the nucleus but also in the cytoplasm, where it may function as a post-transcriptional regulator.]

The estrogen receptor (ER) plays a crucial role not only in the nucleus but also in the cytoplasm, where it functions as a post-transcriptional regulator. The estrogen receptor has an important function as the transcriptional and post-transcriptional regulation of cellular metabolism.

[2. The author should highlight the significance of selective estrogen receptor degraders (SERDs) in the context of heterodimerization, particularly addressing the impact of ER-alpha heterodimerization with other proteins. This would provide a more comprehensive understanding of the mechanisms underlying the therapeutic efficacy of SERDs.]

SERDs are a drug class that targets estrogen receptor alpha (ERα) for proteasome-dependent degradation. SERDs have the ability to induce receptor degradation through creating an unstable protein complex, which may overcome mechanisms of resistance to aromatase inhibitors and selective estrogen receptor modulators (SERMs). SERDs target on ERα for proteasome-dependent degradation has a potential impact on ERα heterodimerization with other proteins (6).

Round 2

Reviewer 1 Report

Comments and Suggestions for Authors

 Request  Satisfied.  Hence, for me :OK for publication

Reviewer 2 Report

Comments and Suggestions for Authors

The authors have addressed all the comments. 

Comments on the Quality of English Language

Moderate editing of the English language required